# Gibbs process distinguishes survival and reveals contact-inhibition genes in Glioblastoma multiforme

Afrooz Jahedi[1☯], Gayatri Kumar[1☯], Lavanya Kannan[2], Tarjani Agarwal[3], Jason Huse[1,4], Krishna Bhat[1,5], Kasthuri Kannan[1,5]*

**1** Department of Translational Molecular Pathology, UT MD Anderson Cancer Center, Houston, TX, United States of America, **2** Ernst & Young, Houston, TX, United States of America, **3** Point of Care, Livingston, NJ, United States of America, **4** Department of Pathology, UT MD Anderson Cancer Center, Houston, TX, United States of America, **5** Department of Neurosurgery, UT MD Anderson Cancer Center, Houston, TX, United States of America

☯ These authors contributed equally to this work.
* kskannan@mdanderson.org

**Data Availability Statement:** The downloaded TCGA images can be viewed under the following link: https://portal.gdc.cancer.gov/image-viewer?filters=%7B%22op%22%3A%22AND%22%2C%

## Abstract

Tumor growth is a spatiotemporal birth-and-death process with loss of heterotypic contact-inhibition of locomotion (CIL) of tumor cells promoting invasion and metastasis. Therefore, representing tumor cells as two-dimensional points, we can expect the tumor tissues in histology slides to reflect realizations of spatial birth-and-death process which can be mathematically modeled to reveal molecular mechanisms of CIL, provided the mathematics models the inhibitory interactions. Gibbs process as an inhibitory point process is a natural choice since it is an equilibrium process of the spatial birth-and-death process. That is if the tumor cells maintain homotypic contact inhibition, the spatial distributions of tumor cells will result in Gibbs hard core process over long time scales. In order to verify if this is the case, we applied the Gibbs process to 411 TCGA Glioblastoma multiforme patient images. Our imaging dataset included all cases for which diagnostic slide images were available. The model revealed two groups of patients, one of which - the "Gibbs group," showed the convergence of the Gibbs process with significant survival difference. Further smoothing the discretized (and noisy) inhibition metric, for both increasing and randomized survival time, we found a significant association of the patients in the Gibbs group with increasing survival time. The mean inhibition metric also revealed the point at which the homotypic CIL establishes in tumor cells. Besides, RNAseq analysis between patients with loss of heterotypic CIL and intact homotypic CIL in the Gibbs group unveiled cell movement gene signatures and differences in Actin cytoskeleton and RhoA signaling pathways as key molecular alterations. These genes and pathways have established roles in CIL. Taken together, our integrated analysis of patient images and RNAseq data provides for the first time a mathematical basis for CIL in tumors, explains survival as well as uncovers the underlying molecular landscape for this key tumor invasion and metastatic phenomenon.

22content%22%3A%5B%7B%22op%22%3A%
22in%22%2C%22content%22%3A%7B%22field
%22%3A%22cases.case_id%22%2C%22value%
22%3A%5B%22set_id%
3AopEizoQB3nUOJWKe2uR_%22%5D%7D%7D
%5D%7D Codes and data for this project can be
found at: https://github.com/kannan-kasthuri/
kannan-kasthuri.github.io/tree/master/research/
Gibbs

**Funding:** The author(s) received no specific
funding for this work.

**Competing interests:** The authors have declared
that no competing interests exist.

# Introduction and preliminaries

Contact-inhibition of locomotion (CIL) is a fundamental mechanism that immobilizes cells in healthy tissues [1] and has long been implicated in the invasion of cancer cells [2]. When collisions happen between the same cell type, this mechanism is referred to as homotypic CIL, and heterotypic CIL occurs when the collisions are between different cell types. Several studies have demonstrated a loss of heterotypic CIL between tumor and normal cells and have hypothesized this as a phenomenon behind tumor invasion and metastasis [3–5]. At the same time, it is known that homotypic CIL is maintained between the tumor cells [6, 7]. Hence, we can expect the spatial distribution of cells in the tumor tissue to reflect one or both of these phenotypes that would distinguish the survival of the patients, with loss of heterotypic CIL patients evidencing low survival due to tumor invasion and metastasis.

Although CIL is a dynamic mechanism that happens in real-time, by examining the spatial distribution of cells in tumor tissues, we can infer homotypic CIL and/or loss of heterotypic CIL. After all, if homotypic inhibition is maintained, the tumor cells should not be too clustered. Similarly, if the loss of heterotypic inhibition occurs, we can expect a clustering phenotype. Even though one cannot infer the mechanisms of homotypic or loss of heterotypic CIL based on dispersion or clustering phenotypes alone, we can deduce these processes if cell movement gene expression is also altered. Therefore, by examining the spatial statistics and applying appropriate inhibitory or cluster models, homotypic CIL or loss of heterotypic CIL can be inferred by integrating pathology images with gene expression data. Even though CIL as a collective migration of tissues has been a subject of extensive mathematical modeling [8], application of the stochastic process to understand these mechanisms in tumors through radiomics approaches has not been undertaken, and the mathematical description of CIL as a spatial point process remains elusive.

From a mathematical standpoint, several stochastic models for tumor evolution has been proposed since 1950's. Five dominant paradigms have emerged [9], out of which Two-Stage Clonal Expansion (TSCE) and Moran models have been predominantly applied to study tumor growth [10–12]. TSCE posits that tumor growth follows two stages, an initiating stage which is modeled as an inhomogeneous Poisson process (because tumor initiation is a rare event) and proliferation/malignancy stage that follows the birth-and-death process. Thus, the Moran model also being a birth- and-death process, allows us to infer tumor growth is a birth-and-death process with CIL as a critical mechanism for invasion and metastasis.

Therefore, representing tumor cells as two-dimensional points, we can expect the tumor tissues in histology slides to reflect realizations of spatial birth-and-death process where each point/cell would maintain a certain distance from its neighbors if homotypic CIL is maintained. These spatial points can be mathematically modeled using an appropriate spatial point process that considers inhibitory interactions, and Gibbs hard core process as an inhibitory process to which birth-and-death process converges is a natural choice. A thorough treatment of spatial point processes and in particular Gibbs process is mathematically sophisticated, and beyond the scope of this article. However, we will illustrate the central ideas behind Gibbs process and related statistics transliterated in the language of cells and CIL. These are adapted (sometimes ad verbatim) from excellent introductory point process resources such as [13] and transliterated for our purposes.

## Overview of the Gibbs process and L-function (transliterated to reflect biological context)

We will use the word intensity to refer to the number of cells per unit area in a randomly chosen small region of a histology slide.

**Gibbs hard core process.** A spatial distribution of a finite collection of cells with intensity β can be assigned a probability density with respect to a completely random collection of cells with unit intensity. This probability density will change depending on the interaction between the cells. In the case of intact homotypic CIL in tumors, the cells are forbidden to come too close to each other, and when they maintain a mean distance h, the locations of the centers of the cells form a point process to which each pair of neighboring points is approximately h units apart. Such inhibition is called a hard-core constraint, and the Gibbs hard core process is defined by the probability density when it satisfies the hard-core constraint.

It is noteworthy to mention that there is a deep relationship between the spatial birth-and-death process and the Gibbs hard core process. Preston [14] proves, under the condition that there is a certain minimal distance maintained between the cells, the spatial birth-and-death process will converge to a realization of the Gibbs hard core process. In particular, we have the following result:

Suppose that, in each small interval of time Δt, each existing cell has probability d(t)Δt of undergoing cell death, where d(t) is the apoptosis rate per cell per unit time. In the same time interval, in any small region of area Δa, let a cell division take place with probability m(t)ΔaΔt, where m(t) is the mitotic rate of the cell per unit time. Provided the dividing cell lies h units away from its nearest neighbor, no matter what the initial state of the cells are, over long time scales, this spatial birth-and-death process would reach an equilibrium in which any snapshot of the cells will be a realization of the Gibbs hard core process with Gibbs process intensity β=m(t)/d(t) and hard-core constraint h.

We can now see that when homotypic CIL is maintained in tumors, the hard-core constraint is naturally satisfied and therefore if we represent the cells as two-dimensional points (segmented from histology slides), the Gibbs process is instinctive to apply. However, we do not know *apriori* if homotypic CIL is maintained for all tumors and we can expect some of the tumors to exhibit loss of heterotypic CIL, and some homotypic CIL. The validity of the Gibbs model as an inhibitory process can be applied to test and distinguish the maintenance of homotypic CIL versus the loss of heterotypic CIL. Moreover, the validity of the model can be determined by the statistical theory that suggests the inference for the Gibbs models should be based on summary statistics such as the L-function [13].

**L-function.** The L-function is the commonly used transformation of the K-function. K-function is one of the very popular statistics for comparing the spatial correlations within point patterns [13] proposed by Ripley [15]. The K-function, $K_r$, of a randomly chosen small region of a histology slide image is the expected number of neighbors within radius r of a typical cell at location u, divided by the intensity β. Note that if we do not divide by the intensity of β, the expected number of neighbors within radius r will vary depending on the scale of the region. That is, if the slides are zoomed out/in, the expected number of r-neighbors will change. Dividing by the intensity β assures us the K-function is independent of scaling operations.

Mathematically,

$$K_r = \frac{1}{\beta} E[\text{number of r} - \text{neighbors of a cell at location u}|\text{there is a cell at } u]$$

for any $r \geq 0$ and any location $u$. The theoretical $K$-function of a homogeneous distribution of cells (aka homogeneous Poisson process) is $\pi r^2$ and this quantity can be used to determine if the cells are clustered or inhibited. For such a comparison, the transformed $K$-function,

namely the *L*-function,

$$L_r = \sqrt{\frac{K_r}{\pi}}$$

can be compared to the L-function of the homogeneous Poisson process, namely, $L_r^{pois} = r$. If $L_r > L_r^{pois}$ the cells would exhibit a clustering phenotype and $L_r < L_r^{pois}$, the cells will be contact-inhibited.

As noted above, if the cells are contact-inhibited, the validity of the Gibbs process can be determined by the summary statistics, the L-function. That is, if the cells are contact-inhibited, we can expect the fitted optimal value of Gibbs hard-core constraint, $opt_r G_r$ to be approximately the same as the optimal value of $|L_r - r|$, indicating that the statistics is consistent with the model as determined by the L-function. Therefore, we will use these metrics for our results, that is $L_r \equiv opt_r |L_r - r|$ and $G_r = opt_r G_r$. Thus, it is natural to study the metric $M(r) = (L_r, G_r)$ to determine the convergence of the Gibbs process. We will refer $M(r)$ as the inhibition metric. Further, for our analysis, we will use the smoothed one-dimensional version of the metric, which we call the mean inhibition metric, $MIM(r)$, is defined as:

$$MIM(r) = \frac{L_r + G_r}{2},$$

which we will use to analyze overall survival of the patients.

## Image processing

Fig 1 shows the image processing protocol employed to extract the point pattern to fit the Gibbs process for the TCGA images (image TCGA-02-0339 shown for illustration). We processed all images for which diagnostic slides were available (389 images, see 'S1 File'). We also added additional 22 cases to test the association of MGMT-promotor methylation found to be associated with one of the groups (eventually a false association) in the initial 389 cohort, resulting in 411 patient images. These images were added since there were only a few MGMT-promoter methylated cases in the original 389 cohort that showed a positive association with one of the groups. Three slides, namely, TCGA-19-1388, TCGA-19- 0963 and TCGA-19-1389 had bad diagnostics images, and so we used tissue slides. Each image was resolved until 50 μm and a region of 300 μm$^2$ was randomly chosen and extracted for segmenting the cells.

We note that the downstream analysis is independent of the resolution and the area since Ripley's K-function (and hence the estimated L-function) gives a scale-free description of the expected number of cells within a neighborhood of an arbitrary cell in any region of interest [16]. The cells in the isolated regions were segmented using k-means segmentation algorithm; the center of masses was identified in the binary mask to generate a spatial point pattern corresponding to the images. The standard Gibbs hard core process allows two major generalizations: the Strauss process and the Geyer saturation process. Unlike a Gibbs process, in Strauss process, close pairs of points are not impossible, but are unlikely to occur as a probability. Geyer saturation process is a modification of the Strauss process in which the total number of points from each point is trimmed to never exceed a threshold, also called the saturation parameter (please refer to Chapter 13 in [13]). We fitted the Geyer saturation process, to the point pattern for downstream analysis (refer to 'S2 File' for the type of Gibbs model that was applied). The model parameters and the L-function statistics are presented in S1 Data.

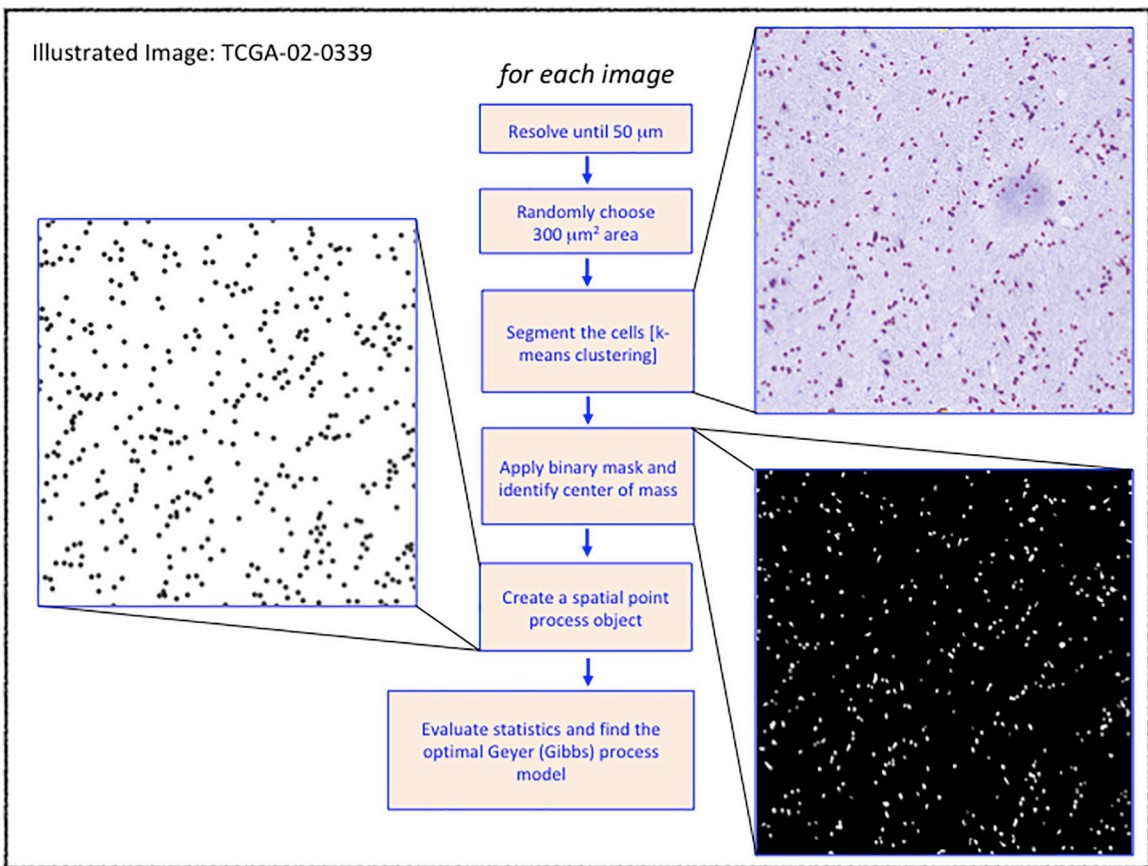

**Fig 1. Image processing protocol.**

## Results

Here we discuss the results after we obtained the $G_r$ and $L_r$ values by processing all the images using the image processing protocol described above. Both $L_r$ and $G_r$ values were scaled using the standardized $z$-score for the rest of the analysis.

### Convergence of the Gibbs process

As noted above, if the Gibbs model fits the data, we would expect the $L_r$ and $G_r$ values to be close and hence the metric M(r) = ($G_r$, $L_r$) is likely to be grouped for some patients. We determined the optimal number of groups for this metric. Fig 2(A) shows the result of applying 30 indices for determining the optimal number of groups (see Methods section for the algorithm used). A majority of the indices reported the optimal number of groups as 2 for this metric. One of the groups, we call the Invasion group, had a smaller range of values for this metric than the other group which we refer as the "Gibbs group."

Fig 2(B) shows the metric M (r) for the Invasion group (red) and the Gibbs group (blue) where 145 patients were in the Invasion group and 266 belonged to the Gibbs group (see S2 Data for group association). We note that the variance of M (r) in the Gibbs group is lesser than in the Invasion group, indicating that $G_r$ and $L_r$ values converge at least for the Gibbs group.

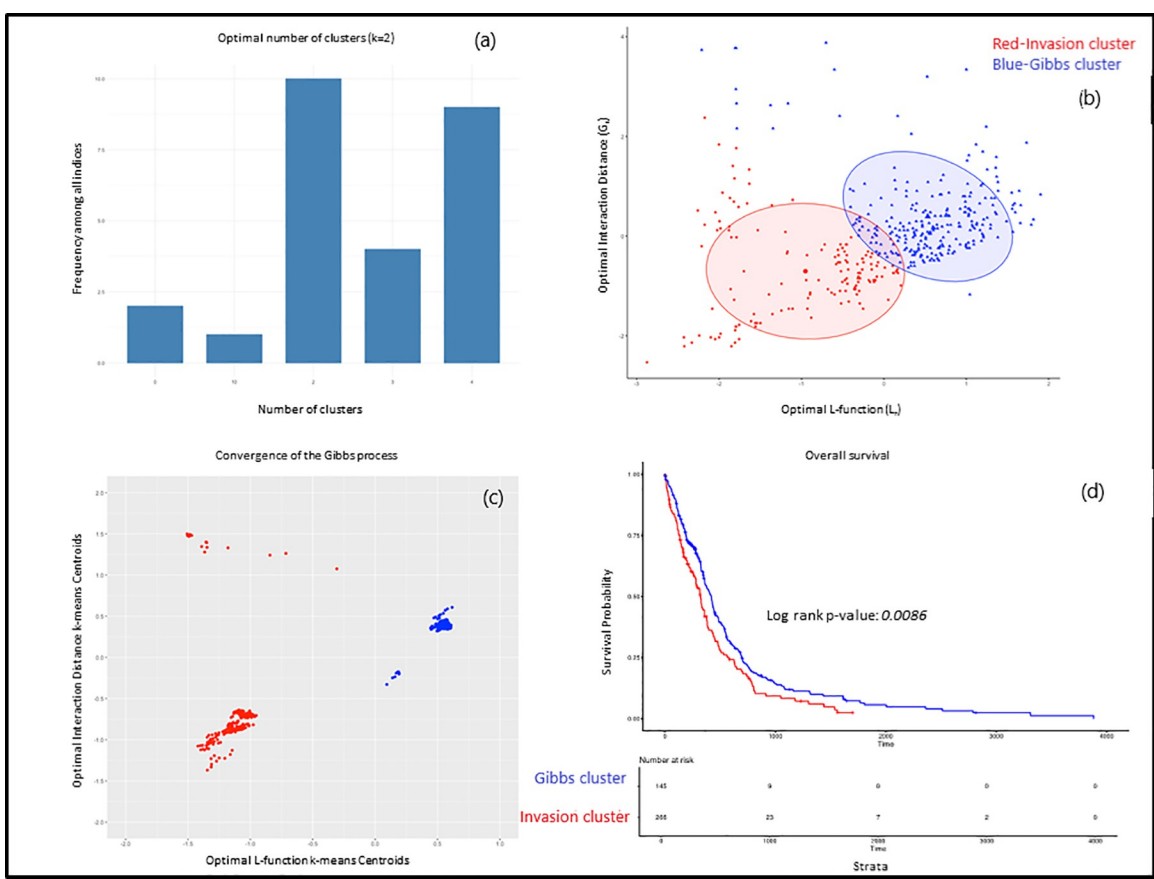

**Fig 2. Convergence of the Gibbs process.**

Therefore, in order to determine if this is a convergence behavior in mathematical terms, we plotted the k-means centroids as we included more M (r) values. Fig 2(C) shows the k-means centroids starting with 20 patients and increasing to 411 patients (see S4 Data for the centroids of 392 patients). We see the two subgroups of patients, one with "clustering" phenotype (Invasion group) and the other "inhibitory" phenotype (Gibbs group). Moreover, calculating the overall survival time for these two groups of patients from the TCGA database, revealed the patients with the clustering phenotype survive significantly shorter than the patients with inhibitory phenotype. This is illustrated in Fig 2(D) (also see S5 Data for the actual numbers). One can expect this result since the patients with loss of heterotypic CIL between tumor and normal cells that result in tumor cell invasion and metastasis are likely to have clustering phenotype and poor survival. On the other hand, patients with homotypic CIL between tumor cells, where cells repel each other, are less prone to tumor cell invasion leading to high survival.

## Gibbs process distinguishes survival time

Although the difference in the overall survival of the patients is significant between the two groups, we found several patients in the Gibbs group with poor survival and patients in the Invasion group with high survival. This finding is contrary to the expectation that if the loss of heterotypic CIL and homotypic CIL distinguishes these patient types, theoretically, the number

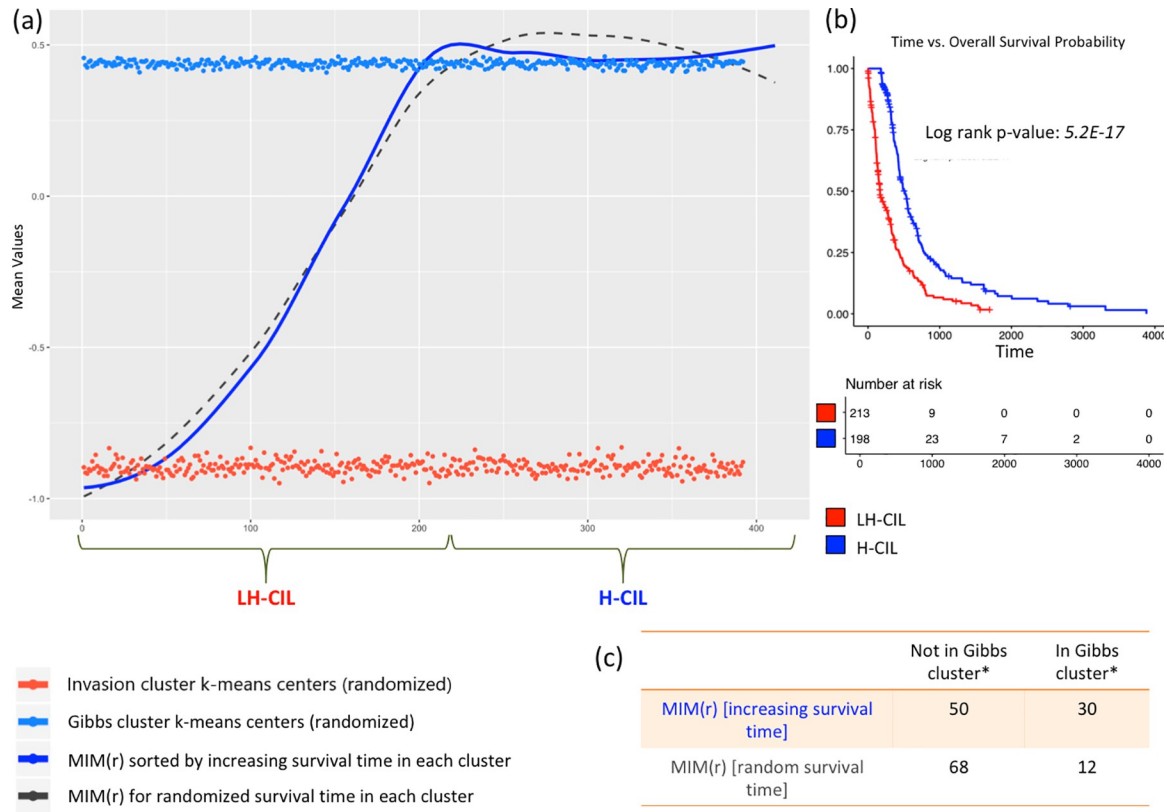

(a)

(b) Time vs. Overall Survival Probability

Log rank p-value: *5.2E-17*

Number at risk

LH-CIL
H-CIL

Invasion cluster k-means centers (randomized)

Gibbs cluster k-means centers (randomized)

MIM(r) sorted by increasing survival time in each cluster

MIM(r) for randomized survival time in each cluster

* Measured up to 4 standard deviations from Gibbs cluster k-means centers (randomized)

(c)

| | Not in Gibbs cluster* | In Gibbs cluster* |
|---|---|---|
| MIM(r) [increasing survival time] | 50 | 30 |
| MIM(r) [random survival time] | 68 | 12 |

Fisher's Exact Test p-value: 0.002

**Fig 3. Survival time analysis.**

of patients in the Gibbs group who do worse should be very small and correspondingly, the number of patients in the Invasion group who survive high should be negligible as well.

In other words, the difference in mean survival between the groups must be much higher (with much lower statistical significance) than what it is currently presented. We hypothesized that this could be due to the noise in the metric $M(r)$ and smoothing the metric according to increasing survival time in each group will likely delineate and identify the mark where actual homotypic CIL is established in patients. Fig 3(A) shows the one dimensional version of the smoothed metric $M(r)$, i.e., $MIM(r)$, using the Loess regression fitting for the increasing survival time (dark blue), and randomized survival time (dashed-gray) patients, where increasing survival time and randomized survival time are determined in each group (the codes and the data for generating this are also provided). It also shows the randomized $k$-means centroids for the two groups as horizontal dots, red and blue.

We note that MIM(r) with increasing survival time is significantly associated (Fig 3(C), Fisher's exact test p-value = 0.002) with the randomized k-means centroids of the Gibbs group than MIM(r) with randomized survival time, demonstrating that it is precisely the noise in the metric M(r) that included many patients with poor survival time in the Gibbs group and high survival time in the Invasion group. This association was measured up to 4 standard deviations from the randomized k-means centroids of the Gibbs group. Moreover, the association of MIM(r) with increasing survival time also revealed the point at which the homotypic CIL establishes in the tumor cells - around 179 days, corresponding to index 213 in the Gibbs group (see S1 Fig and S6 Data). Using this time-point to distinguish loss of heterotypic CIL

(LH-CIL) and homotypic CIL (H-CIL), we see that patients with LH-CIL have an extremely poor outcome than H-CIL patients as shown in Fig 3(B).

## Gene expression signatures between loss of heterotypic CIL and homotypic CIL subgroups

In order to determine if there are expression differences between LH-CIL and H-CIL subgroups, we performed RNA-seq analysis from the TCGA data that was available in the GDC database for several of the patients. There were 22 patients in the Invasion group and 47 patients in the Gibbs group who had the corresponding RNA-seq data (see S7 and S8 Data). The analysis revealed 447 differentially expressed molecules several of which were significantly implicated in cell movement, cell adhesion, and binding (see S9 Data). More specifically, 48 out of the 86 genes involved in cell movement were found to have low enrichment in H-CIL group (green) as compared to LH-CIL group (red) and exhibited decreased cell movement consistent with several findings [17, 18] that show these genes increase cell migration and invasion (see S2–S4 Figs and S10 Data). This can be directly attributed to homotypic CIL that results in inhibition between the tumor cells. Fig 4(A) shows the differential gene expression for cell movement genes with an absolute value of the average gene expression difference between LH-CIL and H-CIL groups greater than 0.2. S7 Fig and S11 Data shows the heatmap and gene expression values (original and transformed) for the signature derived in Fig 4(A). Correspondingly, the increased expression of these 48 cell movement genes in LH-CIL patients demonstrate loss of heterotypic CIL leading to tumor invasion, and hence poor survival.

Further, pathway analysis using the differentially expressed genes revealed genes in Actin cytoskeleton and RhoA signaling as key molecular alterations (Fig 4(B) and 4(C)). S12–S15 Data provides the values of these gene expression signatures and l S5 and S6 Figs shows the implicated signaling pathways.

The actin cytoskeleton plays a critical role in cell movement as they link to extracellular matrix proteins (through adhesions) when cell protrusions stabilize during movement [19]. Also, it is known that the three Rho isoforms – *RhoA, RhoB, and RhoC* – can induce stress fiber formation [7, 20]. Since directional cell migration involves the polarized formation of cell protrusions in the front and contraction of stress fibers at the edge, our results suggest that unstable protrusions and/or deformed stress fibers in LH-CIL patients are likely to induce tumor cell invasion. On the other hand, over-stabilized protrusions and/or formation of additional stress fibers may result in an enhanced CIL in homotypic patients.

## Discussion

The importance of CIL in maintaining normal physiological function has been studied extensively [1, 2] and the loss of CIL is an important contributing factor towards metastasis [1, 4, 5, 21]. CIL is mediated by changes in the cytoskeleton using molecular mechanisms consisting of various process [22–24]. The formation of a physical contact in the form of cell-cell adhesions between the interacting cells is a requirement for CIL [25]. This is mediated by Cadherins, a transmembrane glycoprotein which forms the cell-cell junctions and is a key regulator of the actin cytoskeleton [26]. The Eph receptor which belongs to the tyrosine kinase family also plays a role in cell-cell contact by binding the ephrin ligand on the neighboring cell and coupling the them [27]. The Rho family of GTPases controls cytoskeleton arrangement and dynamics [28] which is crucial in inhibiting protrusion of the leading edge upon cell-cell contact [29]. The Rac1 activity at the leading edge promotes actin polymerization which drives the protrusion [30]. RhoA dependent ROCK activity drives contraction of lamellae and ensures

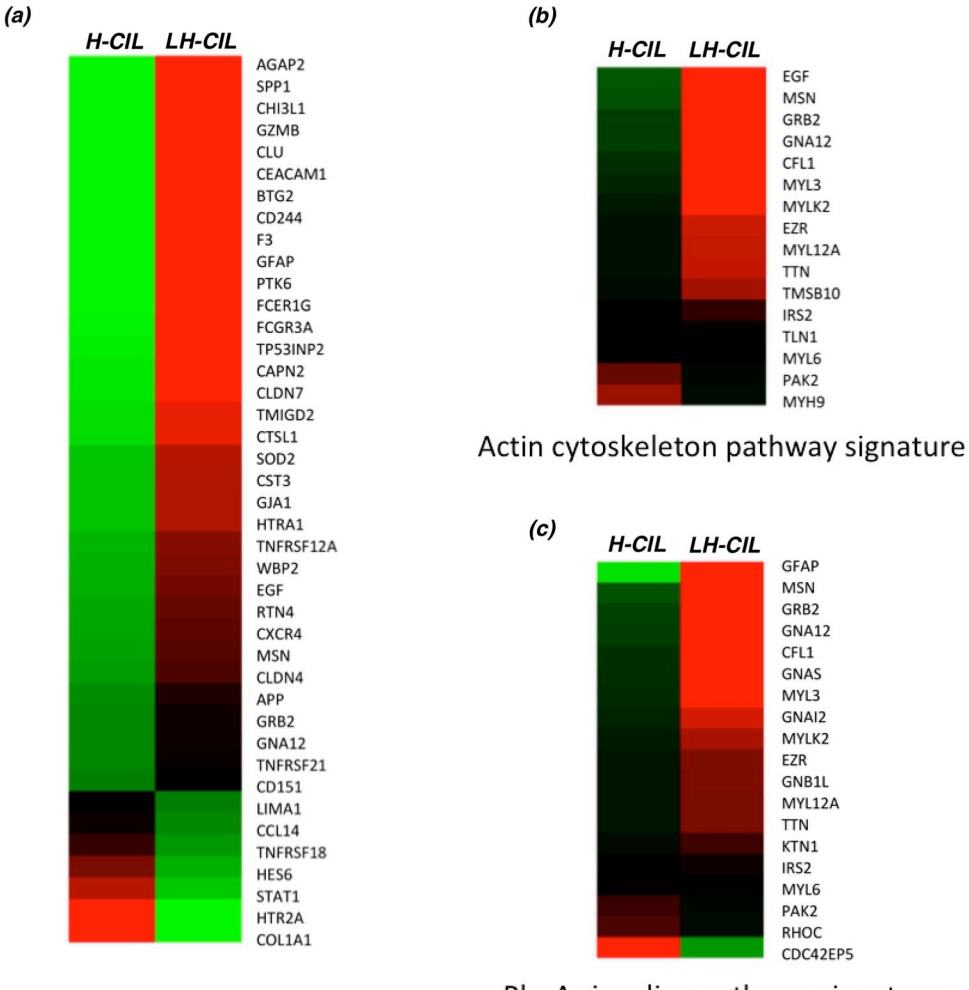

**Fig 4. Gene expression signatures.**

normal CIL function [31–33]. Essentially, RhoA and Rac1 function are reciprocal with one inhibiting the other [34].

Molecular inhibition of N-cadherins also results in protrusive activity by Rac1 at the contact site in neural crest cells leading to loss of CIL [35]. Also, the overexpression of E-cadherins in neural crest cells leads to increase in Rac1 and an eventual loss of CIL [36]. There is evidence that shows that the repolarization of cells by activation of Rac1 in the free edge can lead to separation of the colliding cells. Several events in concert can promote cell separation upon collision. The disassembly of the adhesion complex formed at the contact junction can separate the cells through uncoupling [23]. The buildup in tension upon cell contact leading to an increase in microtubule dynamics mediated by RhoA/ROCK activity can promote separation [22]. The linking of actin retrograde flow in the coupled cells during contact can also lead to tension which is sufficient to pull the cells apart [23, 24, 37]. Together these events can mediate disassembly of cell-cell adhesions and promote cell separation restoring CIL.

CIL in cancers have been extensively studied, especially from a biophysical and molecular perspective [36]. CIL has been shown to be equally important for developmental processes as well as in cancer disease pathology. Previous studies have outlined several stages of CIL including cell-

cell contact, protrusion inhibition and cycles of repolarization and contraction [38]. These processes are regulated by epithelial to mesenchymal transition (a key cancer initiation mechanism) and cadherin switching from dependence on E-cadherin to N-cadherin [39, 40]. These adhesion proteins then signal toward small GTPases to promote or oppose migration [41, 42]. Cell surface receptors such as Ephrin B or Robo are also actively involved in crosstalk with cadherin junction proteins in regulating CIL [43, 44]. Thus, CIL requires coordinated mechanotransduction and is linked to key molecular pathways that regulate cancer progression. In this study, we used mathematical modeling on histology images of glioblastoma patients to link spatial point process to CIL. We provide further validation of this approach by identifying key signaling pathways involved in CIL such as actin cytoskeleton and RhoA signaling as being differential between loss of heterotypic CIL and intact homotypic CIL. In [7], Mayor and Carmona-Fontaine proposes "the invasive behavior of tumors is facilitated by the absence of heterotypic CIL with normal cells, whereas homotypic CIL between cancer cells helps collective migration and/or dispersion of the tumor." Our model suggests this hypothesis is likely to be true.

Spatial point processes are powerful mathematical frameworks for studying point patterns. Cells invariably move, and by representing them as points, it is appropriate to study these movement behaviors using point processes. Gibbs process as an inhibition model is a natural choice for studying inhibitory interactions that drive CIL in tumors. Although there have been efforts to learn tumor's spatial architecture purely from a stochastic process perspective and point processes has been applied in various cancer studies [45–48], to the best of our knowledge, a point process description of CIL has not been attempted. The power of our approach is that these mechanisms can be determined only using microscopy images along with a robust mathematical model. In particular, the fact that tumor cells invade, and this invasion can be deciphered through images presents a dominant approach for cancer diagnostics and treatment options. It opens a window for a plethora of investigations, both clinical and biological, that can be followed when integrated with the existing knowledge on the molecular landscape of Glioblastoma multiforme. Such integration can be made using the marked spatial point processes [49] such as marked Gibbs or marked Geyer saturation process.

This methodology that demonstrates the effectiveness of spatial point pattern analysis in Glioblastoma multiforme, can be applied to any cancer types. Gibbs and Strauss's models are applicable where cell-to-cell 'repulsion' interactions define heterogeneity like in diffuse gliomas. In tumors of epithelial origin, tumor microenvironment support clustering of cells in a process called "nesting." For these tumors or even if there is clustering in diffuse gliomas, probability models such as *Cox*, *Neyman-Scott*, *Matérn*, *Thomas*, *Gauss-Poisson*, *Cauchy* and several other processes can be applied and compared. Moreover, cluster point processes are natural to apply and compare in the setting of clonal evolution since these processes model parent offspring associations that mimic the natural process of cell division. In short, spatial point process models help establishing a strong association between biology and contemporary radiomic approaches in medicine where the primary goal is computing the correlations between computational features and clinical outcomes. Without establishing such associations, translating the results of radiomics into a clinical setting is likely to be unachievable [50], and by integrating spatial point process models, effective radiomic systems can be developed and implemented to translate biological findings to clinical care.

## Methods

### Diagnostic slide images

We queried the GDC portal (https://portal.gdc.cancer.gov/) for GBM cases for which diagnostic cases were available. The query is presented in Supplemental Information.

## Image and data processing

Microscopy H&E images were resolved until 50 μm and a region of 300 μm$^2$ was randomly chosen using QuPath [51], an open source software for digital pathology image analysis. Image segmentation was done using color-based segmentation algorithm that uses k-means clustering in Matlab [52]. Spatial statistics such as the L-function and the parameters of the Geyer saturation process (a Gibbs model) were obtained using spatstat [13], an open source software for spatial statistics.

For determining the optimal number of groups, we used NbClust [53] algorithm that applies 30 indices to estimate the optimum. The metrics for our data set are given in S3 Data. Also, for the k-means clustering, we used the standard package k-means as implemented in the R programming language.

## Gene expression and pathway analysis

RNA-seq analysis was performed using TCGAbiolinks [54] queries which facilitate the GDC open-access data retrieval to perform standard reproducible differential expression analysis. TCGAbiolinks uses EdgeR differential expression package [55] under the query TCGAanalyze_DEA. We used false discovery rate cutoff of 0.001 and log-fold change cutoff of 6 with the generalized linear model glmLRT as parameters to TCGAanalyze_DEA. This resulted in 447 differentially expressed genes (see Supplemental Information). This differential gene expression signature was used in Ingenuity Pathway Analysis (IPA) tool [56] as well as the open-source software ToppGene suite [57], both of which revealed cell movement and adhesion signatures as top Gene Ontology process. Furthermore, IPA analysis revealed actin cytoskeleton and RhoA signaling as critical pathways.

For deriving the cell movement, Actin cytoskeleton, and RhoA signaling signatures, we extracted the differentially expressed genes in these processes and pathways from IPA and used the raw RNA-seq counts to get the average expression. The values were RMS normalized across LH-CIL and H-CIL values and log$_2$ transformed and plotted using the MeV software [58].

## Supporting information

**S1 File. Contains the original query used in the GDC portal to download the images.**
(PNG)

**S2 File. Contains a brief description of the Gibbs processes.**
(PDF)

**S1 Data. Geyer model statistics.**
(XLSX)

**S2 Data. Invasion group and Gibbs group association for the patients.**
(XLSX)

**S3 Data. Optimal number of groups indices by NbClust.**
(XLSX)

**S4 Data. Convergence of the Gibbs process - centroids information.**
(XLSX)

**S5 Data. Survival information.**
(XLSX)

**S6 Data. Information on LH-CIL and H-CIL division.**
(XLSX)

**S7 Data. Original RNA-seq counts for H-CIL patients.**
(XLSX)

**S8 Data. Original RNA-seq counts for LH-CIL patients.**
(XLSX)

**S9 Data. Differentially expressed genes between H-CIL and LH-CIL patients.**
(XLSX)

**S10 Data. Implicated cell movement genes.**
(XLS)

**S11 Data. Cell movement genes RNA-seq values (average and transformed).**
(XLSX)

**S12 Data. Implicated actin pathway genes.**
(XLS)

**S13 Data. Actin pathway RNA-seq values (average and transformed).**
(XLSX)

**S14 Data. Implicated RhoA pathway genes.**
(XLS)

**S15 Data. RhoA pathway RNA-seq values (average and transformed).**
(XLSX)

**S1 Fig. Shows time point/indices at which homotypic CIL is established in the Gibbs group.**
(PNG)

**S2 Fig. Decrease in cell movement as revealed by IPA.**
(PNG)

**S3 Fig. Shows 48/86 genes have differential expression consistent with the direction of decrease in cell movement.**
(PNG)

**S4 Fig. Cell movement as the top GO process as revealed by Toppgene, an open source pathway analysis tool.**
(PNG)

**S5 Fig. Altered RhoA signaling pathway.**
(PNG)

**S6 Fig. Altered actin cytoskeleton pathway.**
(PNG)

**S7 Fig. 86 cell movement gene signature heatmap.**
(TIFF)

## Author Contributions

**Conceptualization:** Kasthuri Kannan.

**Data curation:** Kasthuri Kannan.

**Formal analysis:** Gayatri Kumar, Krishna Bhat, Kasthuri Kannan.

**Investigation:** Gayatri Kumar, Tarjani Agarwal, Jason Huse, Krishna Bhat, Kasthuri Kannan.

**Methodology:** Kasthuri Kannan.

**Project administration:** Lavanya Kannan, Kasthuri Kannan.

**Resources:** Kasthuri Kannan.

**Software:** Kasthuri Kannan.

**Supervision:** Krishna Bhat.

**Validation:** Kasthuri Kannan.

**Visualization:** Kasthuri Kannan.

**Writing – original draft:** Kasthuri Kannan.

**Writing – review & editing:** Afrooz Jahedi, Gayatri Kumar, Krishna Bhat, Kasthuri Kannan.

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
