## [Decision Letter · Decision Letter 0]

12 Jul 2022

PONE-D-22-10989Gibbs Process Determines Survival and Reveals Contact Inhibition Genes in Glioblastoma MultiformePLOS ONE

Dear Dr. Kannan,

Thank you for submitting your manuscript to PLOS ONE. After careful consideration, we feel that it has merit but does not fully meet PLOS ONE’s publication criteria as it currently stands. Therefore, we invite you to submit a revised version of the manuscript that addresses the points raised during the review process.  In particular, please reduce redundancy and add more context of this work by comparing it to existing body of literature, and carefully address the concerns about causality statements. 

We look forward to receiving your revised manuscript.

Kind regards,

Yi Jiang

Academic Editor

PLOS ONE

Journal Requirements:

Additional Editor Comments :

Please make sure that you address all the reviewers' comments and critiques.

Reviewers' comments:

Reviewer's Responses to Questions

**Comments to the Author**

1. Is the manuscript technically sound, and do the data support the conclusions?

Reviewer #1: Partly

Reviewer #2: Partly

2. Has the statistical analysis been performed appropriately and rigorously? 

Reviewer #1: Yes

Reviewer #2: N/A

3. Have the authors made all data underlying the findings in their manuscript fully available?

Reviewer #1: Yes

Reviewer #2: Yes

4. Is the manuscript presented in an intelligible fashion and written in standard English?

Reviewer #1: Yes

Reviewer #2: No

5. Review Comments to the Author

Reviewer #1: Comments for Authors PLoS ONE manuscript PONE-D-2210989: “Gibbs process determines survival and reveals contact inhibition genes in glioblastoma multiforme”

General Comments:

1. Many thanks to the authors for an interesting paper linking spatial point processes, tumor biology, and glioblastoma survival. The paper links multiple concepts and methods and remains readable and informative throughout.

2. I suggest adding some text distinguishing between spatial clustering (or inhibition) of the cell centers in the image from the “clusters” arising from the k-means classification. The term “cluster” can refer to aggregations in the image space for the former and in the classification space for the latter. This needs to be clearly stated to motivate the authors’ terms of the “invasion cluster” and the “Gibbs cluster”, since, unless I’m misreading the results, these do not refer to clusters of cells in the images.

3. The title and the conclusion make some strong causal statements that I’m not sure are completely supported by the methods. Specifically, the title notes that a Gibbs process “determines” survival and the last paragraph states “spatial point process models establish the causal relationship between biology and contemporary radiometric approaches in medicine.” The spatial point process method examines the pattern of cells via the L-function but I don’t believe matching the L-function uniquely defines the biological causal mechanism. Two important elements of spatial point process theory are that, without additional information, a single pattern cannot mathematically distinguish between a pattern of independent events (point locations) with an inhomogeneous intensity, from a pattern of dependent events with a homogeneous intensity (Bartlett, 1964, Spectral analysis of two-dimensional point processes. Biometrika. 51, 299-311). That is, the mathematical properties cannot uniquely distinguish between dependence (or inhibition) and intensity. In addition, Baddeley and Silverman (1984, A cautionary example on the use of second-order methods for analyzing point patterns. Biometrics, 40, 1089-1093) give an example of two very different point processes that have identical intensities and K-functions. The “without additional information” component of the Bartlett (1964) results offers some room for hope (e.g. repeated observations where clusters appear in the same locations would offer evidence of an inhomogeneous intensity, while similar clusters in differing locations over repeated observations would suggest dependences between point events. That said, while the quantification of the L function and matching it to that of a Gibbs process offers evidence of association, I don’t feel it can be viewed as a causal inference result. I suggest revising the statements accordingly.

Specific Comments:

1. Line 39. “process in which” to “process to which”?

2. Line 58. “which each pair of points” to “which each pair of neighboring points”? (I believe the inhibition distance refers to the nearest neighbors, not *all* pairs of points).

3. Line 62. “a certain distance” to “a certain minimal distance”.

4. Line 78. “(or both)”. Does “both” refer to both H-CIL and LH-CIL?

5. Line 85. “analyzing” to “comparing”? Also, should “correlations between point patterns” be “correlations within point patterns”? I believe the K function summarizes correlations within a pattern, and comparing K functions between two point patterns is comparing this within-pattern correlation rather than summarizing a correlation between the two processes.

6. Line 94. “tested against” to “compared to”.

7. Line 100. It is unclear to me what is meant by “the model fits the statistics”, perhaps “the statistics are consistent with the model”?

8. Lines 112-113. “to further test the association of MGMT-promoter…”. This sentence is somewhat confusing. It suggests adding 22 cases to test an association that is then dismissed as a false association. The motivation for including the cases needs to be more clearly defined. It currently sounds as if they were added to test for an effect that was later dismissed.

9. Line 132. I suggest adding a reference to (or definition of) the Geyer saturation process.

10. Line 158. “we” to “We”.

11. Line 163. “to clustered” to “to be clustered”.

12. Line 180. “significantly shorter survival durations” to “significantly shorter survival”?

13. Line 189-190. “had a shorter survival duration” and “had a longer survival duration.” Shorter or longer than what? Readers would benefit from a clearer statement. (Also, is “survival duration” the same as “survival”?)

14. Line 194. “log-rank statistical significance…is much higher”. I’m not sure what “higher statistical significance…than what is currently measured” means here. Can this be clarified for the reader?

15. Line 202. Stray “?”

16. Line 208. “p=0.002 than MIM(r)”?

17. Figure 3 caption. What do the authors mean by “randomized survival duration”? Were survival times randomized to individual cells?

Reviewer #2: Review report on the paper “Gibbs Process Determines Survival and Reveals Contact Inhibition Genes in Glioblastoma Multiforme” with PONE-D-22-10989 submitted to PLOS ONE.

Authors: Kasthuri Kannan et al.

Major Contributions:

It is well known that spatial point process is a mathematical framework for studying cell movement behaviors. In this paper, the authors model the loss of heterotypic contact inhibition of locomotion (CIL) in tumors by using the Gibbs process, which is an equilibrium process of the spatial birth-and-death process. The contribution of this paper is to establish the mathematical foundation of CIL using the popular mathematical theory. The invasive behavior of tumors is facilitated by the absence of heterotypic CIL with normal cells, and the homotypic CIL between cancer cells can help collective migration and/or dispersion of the tumor. The developed methods in this paper open the new window to a plethora of investigations, both clinical and Biological in practice. This paper makes a very nice contribution to the cell movement behavior using applied probability and statistics models.

Major Comments:

1. It is worthwhile to add details of the algorithms for the proposed methods and make the software of the new methods publically available for the use of readers.

2. It is of interest to provide a comparison of the proposed methods with existing ones in the real data analysis.

3. It is worthwhile shorten down the length of the paper since the paper repeated in many places. It is good to make the paper concise in the text as well.

4. It is interesting to add the comparison for the computational cost of the proposed methods with existing ones in real data and simulation studies.

5. There exists grammatical error, typo etc. It would be better to correct all the mistakes. Also see the following minor comments.

Minor Comments:

P. 2, Line 2, ”contact” should be “Contact”

P. 2, Line 9: “in” should be “in the”.

P. 3, line 42, “; however” should be “. However”.

P. 3, line 74, “; therefore” should be “. Therefore”.

P. 4, line 88, “divide” should be “be divided”.

P. 4, line 105, “defined” -> “is defined”.

The above lists are examples until page 4. Similar phenomena appeared in many places.

6. PLOS authors have the option to publish the peer review history of their article (what does this mean?). If published, this will include your full peer review and any attached files.

Reviewer #1: No

Reviewer #2: No

---

## [Author Response · Author response to Decision Letter 0]

25 Aug 2022

Response to the Editor

Editor’s Comment:

“In particular, please reduce redundancy and add more context of this work by comparing it to existing body of literature, and carefully address the concerns about causality statements.”

Corresponding Author’s Response

The entire manuscript is substantially revised to eliminate any redundancy in the statements. We have added more context to the biology of CIL in the discussion section with appropriate references. Also, the manuscript nowhere claims any causality now but highlights the association identified in the study. The title is appropriately changed from “Gibbs Process Determines Survival and Reveals Contact-Inhibition Genes in Glioblastoma Multiforme” to “Gibbs Process Distinguishes Survival and Reveals Contact-Inhibition Genes in Glioblastoma Multiforme” to reflect the association (and not causality). Similar changes to causality statements from the discussion sections are removed and replaced with association. 

Response to the Reviewer 1

General Comments:

1. Many thanks to the authors for an interesting paper linking spatial point processes, tumor biology, and glioblastoma survival. The paper links multiple concepts and methods and remains readable and informative throughout.

Corresponding author response: We thank the reviewer for reviewing our work.

2. I suggest adding some text distinguishing between spatial clustering (or inhibition) of the cell centers in the image from the “clusters” arising from the k-means classification. The term “cluster” can refer to aggregations in the image space for the former and in the classification space for the latter. This needs to be clearly stated to motivate the authors’ terms of the “invasion cluster” and the “Gibbs cluster”, since, unless I’m misreading the results, these do not refer to clusters of cells in the images.

Corresponding author response: We thank the reviewer for pointing this out. Yes, we realized it could be confusing to use the term “cluster” for both patient and spatial aggregation. Therefore, we have changed the term “cluster” to “group” for the patient aggregation. The manuscript is entirely revised with this nomenclature, Gibbs group and Invasion group, and we hope this could make the manuscript more readable.

3. The title and the conclusion make some strong causal statements that I’m not sure are completely supported by the methods. Specifically, the title notes that a Gibbs process “determines” survival and the last paragraph states “spatial point process models establish the causal relationship between biology and contemporary radiometric approaches in medicine.” The spatial point process method examines the pattern of cells via the L-function but I don’t believe matching the L-function uniquely defines the biological causal mechanism. Two important elements of spatial point process theory are that, without additional information, a single pattern cannot mathematically distinguish between a pattern of independent events (point locations) with an inhomogeneous intensity, from a pattern of dependent events with a homogeneous intensity (Bartlett, 1964, Spectral analysis of two-dimensional point processes. Biometrika. 51, 299-311). That is, the mathematical properties cannot uniquely distinguish between dependence (or inhibition) and intensity. In addition, Baddeley and Silverman (1984, A cautionary example on the use of second-order methods for analyzing point patterns. Biometrics, 40, 1089-1093) give an example of two very different point processes that have identical intensities and K-functions. The “without additional information” component of the Bartlett (1964) results offers some room for hope (e.g. repeated observations where clusters appear in the same locations would offer evidence of an inhomogeneous intensity, while similar clusters in differing locations over repeated observations would suggest dependences between point events. That said, while the quantification of the L function and matching it to that of a Gibbs process offers evidence of association, I don’t feel it can be viewed as a causal inference result. I suggest revising the statements accordingly.

Corresponding author response: Again, we thank the reviewer for pointing this out. Yes, the authors realize causality cannot be inferred from models. The original claim for this causality also stemmed from the fact that Gibbs process is the equilibrium process for the spatial birth and death process, and it is established that cells undergo spatial birth and death process. However, looking purely from a spatial context, the reviewer is right, and causality is tough to argue. Therefore, we have completely revised the manuscript and eliminated all claims of causality, including the title change from “Gibbs Process Determines Survival and Reveals Contact-Inhibition Genes in Glioblastoma Multiforme” to “Gibbs Process Distinguishes Survival and Reveals Contact-Inhibition Genes in Glioblastoma Multiforme” to reflect the association (and not causality). 

Specific Comments:

1. Line 39. “process in which” to “process to which”? 

2. Line 58. “which each pair of points” to “which each pair of neighboring points”? (I believe the inhibition distance refers to the nearest neighbors, not *all* pairs of points).

3. Line 62. “a certain distance” to “a certain minimal distance”.

4. Line 78. “(or both)”. Does “both” refer to both H-CIL and LH-CIL?

5. Line 85. “analyzing” to “comparing”? Also, should “correlations between point patterns” be “correlations within point patterns”? I believe the K function summarizes correlations within a pattern, and comparing K functions between two point patterns is comparing this within-pattern correlation rather than summarizing a correlation between the two processes.

6. Line 94. “tested against” to “compared to”.

7. Line 100. It is unclear to me what is meant by “the model fits the statistics”, perhaps “the statistics are consistent with the model”?

8. Lines 112-113. “to further test the association of MGMT-promoter…”. This sentence is somewhat confusing. It suggests adding 22 cases to test an association that is then dismissed as a false association. The motivation for including the cases needs to be more clearly defined. It currently sounds as if they were added to test for an effect that was later dismissed.

9. Line 132. I suggest adding a reference to (or definition of) the Geyer saturation process.

10. Line 158. “we” to “We”.

11. Line 163. “to clustered” to “to be clustered”.

12. Line 180. “significantly shorter survival durations” to “significantly shorter survival”?

13. Line 189-190. “had a shorter survival duration” and “had a longer survival duration.” Shorter or longer than what? Readers would benefit from a clearer statement. (Also, is “survival duration” the same as “survival”?)

14. Line 194. “log-rank statistical significance…is much higher”. I’m not sure what “higher statistical significance…than what is currently measured” means here. Can this be clarified for the reader?

15. Line 202. Stray “?”

16. Line 208. “p=0.002 than MIM(r)”?

17. Figure 3 caption. What do the authors mean by “randomized survival duration”? Were survival times randomized to individual cells?

Corresponding author response: All the typos and errors above have been corrected. Thanks for pointing these out.

Response to the Reviewer 2

Major Contributions:

It is well known that spatial point process is a mathematical framework for studying cell movement behaviors. In this paper, the authors model the loss of heterotypic contact inhibition of locomotion (CIL) in tumors by using the Gibbs process, which is an equilibrium process of the spatial birth-and-death process. The contribution of this paper is to establish the mathematical foundation of CIL using the popular mathematical theory. The invasive behavior of tumors is facilitated by the absence of heterotypic CIL with normal cells, and the homotypic CIL between cancer cells can help collective migration and/or dispersion of the tumor. The developed methods in this paper open the new window to a plethora of investigations, both clinical and Biological in practice. This paper makes a very nice contribution to the cell movement behavior using applied probability and statistics models.

Corresponding author response: We appreciate the reviewer for taking the time to review our manuscript and for identifying the scope. It is gratifying.

Major Comments:

1. It is worthwhile to add details of the algorithms for the proposed methods and make the software of the new methods publically available for the use of readers.

Corresponding author response: All the codes and the data has been uploaded and available publicly. We would suggest the reviewer to look into:

https://github.com/kannan-kasthuri/kannan-kasthuri.github.io/tree/master/research/Gibbs

Moreover, the methods that were used are open-source and publicly available, including the spatstat R package that has several models that can be readily used. 

Details of the algorithm can be found in [5] as mentioned in the supplemental as well as in the text, namely, Adrian Baddeley and Rolf Turner. “Spatstat: an R package for analyzing spatial point patterns.” In: Journal of Statistical Software 12.6 (2005). ISSN 1548-7660, pp. 1–42. URL: www.jstatsoft.org.

2. It is of interest to provide a comparison of the proposed methods with existing ones in the real data analysis.

Corresponding author response: The scope of the manuscript is to provide a mathematical basis for CIL and in particular homotypic and heterotypic CIL. Gibbs process has been applied because it is an inhibitory process that confirms to the contact-inhibition. In particular, this is not a methods paper per se. It connects the biology with mathematics and comparison of multiple methods will alter the scope of the manuscript in entirety and will not reflect the biology that the method is designed to model. 

3. It is worthwhile shorten down the length of the paper since the paper repeated in many places. It is good to make the paper concise in the text as well.

Corresponding author response: Yes, the revised version is shortened in the discussion section and does not have any redundancy. We thank the author for this comment.

4. It is interesting to add the comparison for the computational cost of the proposed methods with existing ones in real data and simulation studies.

Corresponding author response: As mentioned in the response for the point no. 2 above, discussing the computational cost and algorithmic details will later the scope of the manuscript substantially. Please refer to the response for comment # 2.

5. There exists grammatical error, typo etc. It would be better to correct all the mistakes. Also see the following minor comments.

Corresponding author response: We thank the reviewer for identifying these typos and errors. They all stand corrected now. 

Minor Comments:

P. 2, Line 2, ”contact” should be “Contact”

P. 2, Line 9: “in” should be “in the”.

P. 3, line 42, “; however” should be “. However”.

P. 3, line 74, “; therefore” should be “. Therefore”.

P. 4, line 88, “divide” should be “be divided”.

P. 4, line 105, “defined” -> “is defined”.

The above lists are examples until page 4. Similar phenomena appeared in many places.

Corresponding author response: All the typos and errors above have been corrected. Thanks for pointing these out.

---

## [Decision Letter · Decision Letter 1]

24 Oct 2022

Gibbs Process Distinguishes Survival and Reveals Contact Inhibition Genes in Glioblastoma Multiforme

PONE-D-22-10989R1

Dear Dr. Kannan,

We’re pleased to inform you that your manuscript has been judged scientifically suitable for publication and will be formally accepted for publication once it meets all outstanding technical requirements.

Kind regards,

Yi Jiang

Academic Editor

PLOS ONE

Additional Editor Comments (optional):

Reviewers' comments:

Reviewer's Responses to Questions

**Comments to the Author**

1. If the authors have adequately addressed your comments raised in a previous round of review and you feel that this manuscript is now acceptable for publication, you may indicate that here to bypass the “Comments to the Author” section, enter your conflict of interest statement in the “Confidential to Editor” section, and submit your "Accept" recommendation.

Reviewer #1: All comments have been addressed

Reviewer #2: (No Response)

2. Is the manuscript technically sound, and do the data support the conclusions?

Reviewer #1: Yes

Reviewer #2: Partly

3. Has the statistical analysis been performed appropriately and rigorously? 

Reviewer #1: Yes

Reviewer #2: N/A

4. Have the authors made all data underlying the findings in their manuscript fully available?

Reviewer #1: Yes

Reviewer #2: Yes

5. Is the manuscript presented in an intelligible fashion and written in standard English?

Reviewer #1: Yes

Reviewer #2: Yes

6. Review Comments to the Author

Reviewer #1: Thank you for your careful and responsive review, all of my original concerns have been adequately addressed.

Reviewer #2: Review report on the paper “Gibbs Process Determines Survival and Reveals Contact Inhibition Genes in Glioblastoma Multiforme” with PONE-D-22-10989R1 submitted to PLOS ONE.

Authors: Kasthuri Kannan et al.

Major Contributions:

It is well known that spatial point process is a mathematical framework for studying cell movement behaviors. In this paper, the authors model the loss of heterotypic contact inhibition of locomotion (CIL) in tumors by using the Gibbs process, which is an equilibrium process of the spatial birth-and-death process. The contribution of this paper is to establish the mathematical foundation of CIL using the popular mathematical theory. The invasive behavior of tumors is facilitated by the absence of heterotypic CIL with normal cells, and the homotypic CIL between cancer cells can help collective migration and/or dispersion of the tumor. The developed methods in this paper open the new window to a plethora of investigations, both clinical and Biological in practice. This paper makes a very nice contribution in the cell movement behavior using applied probability and statistics model. The new version addressed most of my comments. It made an improvement over the original one.

Major Comments:

The journal PLOS ONE is an applied journal. In order to show the benefit and advantage of the proposed method, it is essential to provide a comparison of the proposed methods with existing ones in the real data analysis.

7. PLOS authors have the option to publish the peer review history of their article (what does this mean?). If published, this will include your full peer review and any attached files.

Reviewer #1: No

Reviewer #2: No

---

## [Editor Report · Acceptance letter]

15 Dec 2022

PONE-D-22-10989R1 

Gibbs Process Distinguishes Survival and Reveals Contact-Inhibition Gene in Glioblastoma Multiforme 

Dear Dr. Kannan:

I'm pleased to inform you that your manuscript has been deemed suitable for publication in PLOS ONE. Congratulations! Your manuscript is now with our production department. 

Kind regards, 

on behalf of

Dr. Yi Jiang 

Academic Editor

PLOS ONE